# Modelling, Validation and Experimental Analysis of Diverse RF-MEMS Ohmic Switch Designs in View of Beyond-5G, 6G and Future Networks—Part 1

**DOI:** 10.3390/s23073380

**Published:** 2023-03-23

**Authors:** Jacopo Iannacci

**Affiliations:** Center for Sensors and Devices, Fondazione Bruno Kessler, 38123 Trento, Italy; iannacci@fbk.eu; Tel.: +39-0461-314-441

**Keywords:** 5G, 6G, Beyond-5G (B5G), Future Networks (FN), MEMS, micro-relays, millimeter waves, Radio Frequency (RF), RF-MEMS, switches

## Abstract

The emerging paradigms of Beyond-5G (B5G), 6G and Future Networks (FN), will capsize the current design strategies, leveraging new technologies and unprecedented solutions. Focusing on the telecom segment and on low-complexity Hardware (HW) components, this contribution identifies RF-MEMS, i.e., Radio Frequency (RF) passives in Microsystem (MEMS) technology, as a key-enabler of 6G/FN. This work introduces four design concepts of RF-MEMS series ohmic switches realized in a surface micromachining process. S-parameters (Scattering parameters) are measured and simulated with a Finite Element Method (FEM) tool, in the frequency range from 100 MHz to 110 GHz. Based on such a set of data, three main aspects are covered. First, validation of the FEM-based modelling methodology is carried out. Then, pros and cons in terms of RF characteristics for each design concept are identified and discussed, in view of B5G, 6G and FN applications. Moreover, ad hoc metrics are introduced to better quantify the S-parameters predictive errors of simulated vs. measured data. In particular, the latter items will be further exploited in the second part of this work (to be submitted later), in which a discussion around compact modelling techniques applied to RF-MEMS switching concepts will also be included.

## 1. Introduction

Looking at 2030 and beyond, disruption of paradigms such as 6G (sixth generation of mobile communications) and Future Networks (FNs) will be paramount. On one hand, evolution of services is already ongoing today. Such a trend will progress in the future, leveraging Artificial Intelligence (AI). Diversely, unprecedented innovation is expected for massive exploitation of AI on the network operation plane [1], from the core to the edge. This will lead to the concept of network of networks, with portions of it able to implement self-reactive (highly resilient) functions, thus enabling evolutionary features, while maintaining proper orchestration with the rest of the network, to work as a whole.

The resulting scenario, in terms of key-enabling technologies (KETs), is very intricate. Navigating it can effectively be simple if one reasons in terms of paradigm shifts (PSs) triggered by 6G and FNs, as discussed in [2,3].

Given such a complex scenario, the focus of this work is around low-complexity Hardware (HW) components to be employed in telecommunications and transmission of data. The specific 6G KET here at stake is that of RF-MEMS, i.e., Radio Frequency (RF) passive components realized in MEMS (MicroElectroMechanical-Systems) technology.

Micromachining techniques applied to the realization of RF passives started to be ventured around the mid-1990s, targeting miniaturized planar waveguides, lumped reactive elements and stubs [4,5]. Shortly after, the employment of actual MEMS actuators within waveguide-type RF devices was investigated, enabling reconfigurability of simple and multi-state networks, such as micro-switches [6] and delay lines [7]. From the early 2000s, the scientific literature reported a variety of findings related to RF-MEMS devices, with focus on technology developments, design concepts, modelling and simulation techniques, along with packaging, encapsulation, integration and reliability of such HW items. To this end, a few contributions regarded as pivotal in consolidating RF-MEMS as a self-standing research area are found in [8,9,10,11,12,13,14,15,16,17].

Bearing in mind Beyond-5G (B5G) and 6G applications, a brief state of the art is now reported, focusing on devices working in higher frequency ranges, i.e., millimeter-waves (mm-Waves) and above.

The electrostatic-driven RF-MEMS capacitive switch reported in [18] exhibits insertion loss (S21 when Close) better than −1.5 dB, and isolation (S21 when Open) better than −15 dB, in the 15–110 GHz range. The packaged switch in [19] scores insertion loss better than −2 dB in the 220–280 GHz range when Close (30 V actuation voltage), and isolation around −16 dB when Open, from 240 GHz to 320 GHz.

Stepping to functions other than switching, phase shifters in RF-MEMS technology must be mentioned. The planar solution reported in [20] shows good RF characteristics in the 70–86 GHz range. The concept in [21] relies on a MEMS-based perforated slab, actuated to reconfigure the phase shift, achieving remarkable performance in the 500–600 GHz range.

MEMS-based reconfigurable RF power attenuators are also demonstrated. To this end, [22] reports an in-line 8-bit cascaded concept that exploits series and shunt resistors to attenuate the RF signal, tested up to 110 GHz. The work in [23] discusses a solution featuring two different RF signal path branches, verified up to 20 GHz.

Going back to RF-MEMS micro-relays, they are also studied to realize high-order switching units and matrices. The concept in [24] is conceived as a basic element for various complexity matrices working up to 40 GHz, with isolation better than −20 dB and insertion loss better than −0.3 dB. Diversely, the work in [25] discusses a 12 × 12 switching matrix in which RF-MEMS micro-relays are packaged and integrated within ceramics-based encapsulation.

Wrapping together the discussion, RF-MEMS are expected to enter the set of KETs for 6G and FNs [3], looking at different networks’ segments, e.g., from the edge to the core infrastructure, along with mobile devices. The remarkable characteristics of RF-MEMS in terms of performance and reconfigurability are well known. All of them are constrained to the behavioral features of MEMS-based RF micro-switches.

This work will focus on four designs of electrostatically driven RF-MEMS series ohmic switches. The pursued targets are threefold. First, an extensive validation of the S-parameter simulation methodology based on Finite Element Method (FEM) against experiments is carried out up to 110 GHz. Second, important considerations on the RF performance of the different switch designs, with their pros and cons, are elaborated. Third, ad hoc metrics are introduced to help compare different data and their level of agreement/disagreement in a more quantitative manner. In particular, such metrics are used in this work to support the validation of simulations against measurements. The same indicators will be exploited more extensively in future works, as mentioned below.

The contribution is arranged as follows. After the current Section 1, the next Section 2 introduces the four RF-MEMS micro-switch design concepts. Section 3 reports and discusses the validation of RF characteristics against experiments, performed up to 110 GHz. Section 4 compares the RF characteristics of the four micro-relays. Eventually, Section 5 collects a few conclusive considerations.

This work, despite being self-contained, will be followed by a second part. Part 2 will develop a more extensive exploitation of the ad hoc metrics here introduced, also applied to compact model-based techniques for the fast RF simulation of RF-MEMS devices.

## 2. RF-MEMS Switch Design Concepts

The exploited RF-MEMS technology is available at Fondazione Bruno Kessler (FBK) in Italy, and detailed in [26]. It is based on 6-inch silicon wafers, above which a surface micromachining process featuring eight lithographic steps is performed. The process includes two passivated conductive layers, i.e., polycrystalline-silicon (630 nm-thick) for biasing lines/electrodes and for resistors, and aluminum (630 nm thick) for RF signal underpasses. The structural parts are realized with a 2–5 µm-thick layer of electrodeposited gold, used for planar waveguides, as well as for suspended membranes (air-gaps). MEMS devices are controlled electrostatically, i.e., imposing proper bias [27].

The set of four RF-MEMS series ohmic switches analyzed in this work is now introduced. All the MEMS micro-relays are based on clamped-clamped membranes, transversal over the RF signal line, and framed within Coplanar Waveguide (CPW) structures. This means that the RF line is interrupted underneath the elevated gold. When the MEMS is in its rest position (OFF), the RF switch is Open. Diversely, when the bias imposed to the micro-relay is above the pull-in threshold, the MEMS collapses (ON state), closing the contact between the RF input/out terminations (Close state).

The microphotographs of the four design concepts are shown in Figure 1, where the in-plane size of the CPW structure is indicated. The RF-MEMS switches named Device A (DA) and Device B (DB) in Figure 1a and Figure 1b, respectively, have both a footprint of 1.4 × 1.8 mm^2^. Device C (DC) and Device D (DD) in Figure 1c and Figure 1d, respectively, have an area of 0.7 × 1.3 mm^2^. Pads for imposing the biasing voltage and ground are placed out of the CPW structures, and are not visible in Figure 1. For completeness, some devices show additional pads in the vicinity of the MEMS mechanical anchoring areas. This is because they are equipped with integrated micro-heaters for reliability purposes [28], not used in nor relevant to the purposes of the current study.

Discussing the electromechanical characteristics of the RF-MEMS switches falls out of the scope of this work. However, a brief experimental overview of the pull-in behavior is reported. The in/out current vs. biasing voltage imposed to the RF-MEMS device, i.e., I-V characteristic, is monitored. A small voltage drop, in the range of a few mV, is applied between the input and output RF terminations, while the MEMS floating and the underlying electrodes are subjected to a voltage ramp. As long as the MEMS is not actuated, the in/out resistance is very large (MΩ range), while it drops to a few Ω when pull-in is reached. The I-V characteristic of the switches in Figure 1 is reported in Figure 2. The in/out current steps from 10–20 fA when the MEMS relays are OFF, to 0.2–0.4 mA when they are ON.

The information that can be inferred by the plot bears mainly an indicative value. In fact, the measured samples come from different wafers, processed in a batch for testing and research purposes, rather than for production. Therefore, the MEMS structural gold thickness, along with the amount of residual stress [29], are not strictly controlled. This means that the different pull-in levels in Figure 2 may not accurately correspond to the characteristics of DA to DD samples realized in a standard production batch. What is worth observing is that all the devices actuate in the 45–65 V range, which is aligned with previous work carried out on similar structures [30].

## 3. Experimental Validation of the RF Characteristics

This section enters into the technical details of S-parameter (Scattering parameter) simulation and validation against experiments of the four design concepts in Figure 1. To facilitate the interpretation of the accuracy of simulations, ad hoc metrics are going to be introduced and exploited.

Simulations are performed with the Ansys HFSS Finite Element Method (FEM) commercial tool, relying on full-3D models of RF-MEMS switches. A semi-automated method is exploited for the generation of such structures. Starting from the 2D layout in GDSII format, a technology file is defined in HFSS, for extrusion of the complete 3D model [31]. Given their similarity, only the HFSS model of DC is shown in Figure 3.

Measured S-parameters are collected from RF-MEMS physical samples on a probe station, by means of a Programmable Network Analyzer (PNA) and Ground–Signal–Ground (GSG) probes (150 μm pitch). The delivered power is 0 dBm and the setup is calibrated with the Line-Reflect-Reflect-Match (LRRM) method, from 100 MHz up to 110 GHz.

The comparison of experiments against simulations of the four RF-MEMS designs in Figure 1 will be reported. A preliminary consideration is due around the significantly wide frequency range observed, i.e., up to 110 GHz. This is done to intentionally challenge the FEM tool at the cost of less accurate predictions in some portion of the analyzed span. The counterparty is that of collecting more information, upon which qualitative and quantitative comparative discussion can be built. Of course, it is always possible to pursue the more standardized approach of refining the FEM analysis around a limited portion of the frequency range, e.g., a few GHz wide, aiming at better accuracy.

To build a quantitative discussion around the accuracy of simulations, two ad hoc metrics are defined per each of the four S-parameters, both concerning magnitude and phase. The first is named Percent Magnitude Difference (PMD) and defines the percent error of the simulated magnitude at a certain frequency. The PMD for the S11 at the frequency *f* is expressed as follows:(1)$${\mathrm{PMD}}_{S11}\left(f\right)=\frac{mag\left(S11_{M}\left(f\right)\right)-mag\left(S11_{S}\left(f\right)\right)}{2}*100$$

In Equation (1), $mag\left(S11_{M}\left(f\right)\right)$ and $mag\left(S11_{S}\left(f\right)\right)$ are, respectively, the measured and simulated magnitude of S11 at the frequency *f*. Since RF-MEMS are passive components, the magnitude by definition cannot be greater than 1. Therefore, the maximum error possible is 1 + 1 = 2. Therefore, the difference is normalized with respect to 2, accordingly. In a similar fashion, the Percent Phase Difference (PPD) is expressed as follows:(2)$${\mathrm{PPD}}_{S11}\left(f\right)=\frac{deg\left(S11_{M}\left(f\right)\right)-deg\left(S11_{S}\left(f\right)\right)}{360}*100$$

In Equation (2), $deg\left(S11_{M}\left(f\right)\right)$ and $deg\left(S11_{S}\left(f\right)\right)$ are, respectively, the measured and simulated phase of S11 at the frequency *f*. As before, such a difference is normalized with respect to the largest error possible, which is 360 degrees.

In addition, Table 1 summarizes the acronyms used in the legends and captions of all the following plots.

Only the S11 and S21 parameters will be displayed. In fact, the analyzed RF-MEMS devices are symmetric; therefore, S22 and S12 do not carry additional information, despite the fact that experimental data can be different. To this end, Appendix A collects all the data omitted here. Specific figures and plots in it will be cited in the following. E.g., for the sake of brevity, all the Smith charts discussed in the following pages are uniquely displayed (and referenced) in Appendix A.

Starting from DA (Figure 1a), the S-parameter characteristics of the switch in Open state (i.e., MEMS OFF) are reported in Figure 4, while the complete set of plots is collected in Figure A1. Despite the fact that measurements are quite noisy in some parts, the quantitative characteristics can be easily inferred, spotting, on the other hand, the spikes that can be attributed to noise, which are therefore not relevant to the discussion. At a glance, the prediction of S11 looks less accurate than that of S21. In particular, looking at reflection in Figure 4a, there are two portions of frequency where the disagreement is more marked, i.e., 32–46 GHz and 49–75 GHz. The corresponding PMD of S11 (red trace in Figure 4e), shows the largest prediction errors of 9% and 20%, respectively, in those sub-ranges.

Focusing on isolation in Figure 4b, the accuracy of simulations is more pronounced. The spikes visible in the measured (red) trace above 55 GHz are mainly to be attributed to noise. In turn, PMD of S21 (black trace in Figure 4e) reports an error always falling within the ±5% range.

Concerning the phase of S11 in Figure 4c, the simulation (black trace) anticipates the first phase inversion of about 8 GHz with respect to experiments (red curve). In addition, the simulated phase slope is more pronounced than the measured one. On the other hand, the measured glitches above 80 GHz are not relevant, as they can be attributed in part to noise, in part to the under-sampling of the measured datasets (201 points). Looking at the PPD of reflection (red trace in Figure 4f), it is within 5% until the first inversion, around 61 GHz. Above, since the simulation predicts the transition at lower frequency, the error steps to around 90%, until the measured inversion at 69 GHz. Further above, neglecting the measured glitches, the error is within 8%, while it jumps in the 22–26% range afterwards, because of the slope difference in Figure 4c.

Focusing now on the phase of isolation in Figure 4d, the simulation (black trace) predicts quite accurately the inversion around 38 GHz. This is confirmed by the narrow glitch of the PPD of S21 (black trace in Figure 4f), due to a small transition frequency underestimation of the simulated curve. In details, neglecting such a glitch, the isolation PPD is within less than 3% up to 50 GHz. Then, the rather bumpy behavior of the measured trace, already stressed in Figure 4b, leads to errors fluctuating in the ±50% range, despite being less significant, in view of the noisy measurement. The last phase inversion is anticipated by the simulation, yielding PPD values in the −80–90% range, in the 93–105 GHz range (black trace in Figure 4f).

A few considerations are due around the Smith chart plots of S11 and S21 in Figure A1q and Figure A1r, respectively. All the Smith chart plots shown in Appendix A, are based on the template created by RF Café and available for download in [32]. What was discussed on the measured vs. simulated characteristics of S11 (Figure 4a) and its phase (Figure 4c), along with PMD (red trace in Figure 4e) and PPD (red trace in Figure 4f), can be observed in Figure A1q. In particular, the two sub-ranges (32–46 GHz and 49–75 GHz) in which the simulation underestimates the S11 (Figure 4a), can be easily spotted when looking at the black vs. red trace in Figure A1q, the former (simulation) being closer to the center of the chart with respect to the latter (experiment). Recalling the anticipated S11 phase inversion of simulation vs. experiments (Figure 4c), it reflects in the red and black traces crossing the negative (capacitive) to positive (inductive) sectors of the Smith charts at different points along the horizontal axis in Figure A1q. Rising up in frequency, the measured phase glitch above 80 GHz (red trace in Figure 4c) corresponds to a fleeting crossing to the capacitive sector and back, confirming that it is not relevant. Finally, the more marked slope of the measured phase in Figure 4c, corresponds to an extra rotation of the black trace with respect to the red one, in Figure A1q. Eventually, Figure A1r compares the measured and simulated isolation (S21) on the Smith chart plot. The rotation is predicted with good accuracy, despite some disagreements in the distance from the center are visible. In particular, the noisy behavior of the measured phase (red trace in Figure 4d) gives rise to a rather scattered behavior of the red trace in Figure A1r, mainly close to the center of the chart.

Following a similar arrangement of plots, the measured vs. simulated characteristics of DA in Close state (i.e., MEMS ON) are reported in Figure 5 (full plots in Figure A2).

Concerning the return loss, most of the disagreement is visible above 60 GHz (Figure 5a), yielding a PMD up to 22% (Figure 5e). The prediction of the S11 phase is qualitatively rather accurate up to around 100 GHz (Figure 5c). Looking at PPD in Figure 5f, apart from the phase inversion disagreement in the 28–34 GHz range, it is within ±20%. Above 100 GHz, the simulation misses the second phase inversion. Putting together all these considerations, a certain disagreement, both qualitative and quantitative, is visible in the S11 Smith chart plot in Figure A2q. Focusing on the insertion loss (Figure 5b), the simulated trace is quite accurate. It notably predicts the S21 step around 50 GHz. Instead, the ripples of the measured trace are negligible, as ascribable to noise. The PMD of S21 (black trace in Figure 5e), apart from a couple of spikes, is always within ±5%. Instead, the simulation fails to predict the phase of S21 in a reasonably accurate fashion (Figure 5d). This is particularly evident in the Smith chart plot in Figure A2r. Despite the fact that the measured and simulation phase rotation extents are comparable, the simulation starts with a disagreement of +180 degrees (left instead that right point on the plot horizontal axis). The S21 PPD, conversely, keeps shifting between ±50% error (Figure 5f). In a nutshell, DA is the worst case in terms of simulation accuracy. The next device instead will be the best case. The characteristics of DB in Open state are reported in Figure 6 (full plots in Figure A3).

Observing the reflection (S11) and isolation (S21) in Figure 6a and Figure 6b, respectively, good qualitative and quantitative superposition of the simulated (black) and measured (red) curves is visible. It must also be stressed that measurements, given the wide observed range, are quite noisy. Looking at the corresponding S11 and S21 PMD, red and black trace in Figure 6e, respectively, and neglecting the spikes due to fluctuations of measurements, they always fall within ±5%. Concerning the phase of S11 and S21, in Figure 6c and Figure 6d, respectively, the match of simulations and measurements is rather good as well. The last phase inversion in both the measured S11 and S21 is very noisy. It is suggested to check the measured S22 in Figure A3, which is cleaner. Concerning both PPDs in Figure 6f, apart from the missed phase inversion regions, they range within a few percents in most of the frequency range, with some peaks up to ±15%. Detailed discussion on the Smith chart plots is skipped for brevity. However, it is interesting to observe how the S11 traces’ crossings in the 52–89 GHz range (Figure 6a) reflect in the Smith chart plot in Figure A3q. Stepping now to DB in Close state, data are reported in Figure 7 (full plots in Figure A4).

In this case, there is no need of detailed discussion. Simulations predict quite accurately the return loss and insertion loss, along with the S11 and S21 phase, over the whole frequency range. The S21 PMD (black trace in Figure 7e) is within −2.5% up to 50 GHz, within −5% and 3% up to 90 GHz, and then touches 7.5%. Concerning the S21 PPD (black trace in Figure 7f), apart from the missed inversion frequency sub-ranges, it is always better than 15%.

The characteristics of DC in Open state are reported in Figure 8 (full plots in Figure A5). From now on, the phase plots are omitted for the sake of brevity (still available in Appendix A).

The simulated reflection shows underestimation of the measured S11, increasingly evident above 55 GHz (Figure 8a). This brings to PMD values (red trace in Figure 8c) within ±3% up to 55 GHz, which then rise to 5% up to 80 GHz, and to around 10% above. However, the noisy characteristics of the measured data must be considered. The prediction of isolation (Figure 8b) looks more accurate, with PMD (black trace in Figure 8c) always within a few (plus/minus) percents, with just a −7% peak around 70 GHz. The prediction of S11 and S21 shows rather large imprecision in identifying the phase inversion frequency. Elsewhere, both PPDs in Figure 8d are within 10%. Finally, the Smith chart plots (Figure A5q,r) summarize the just-developed considerations.

The characteristics of DC in Close state are reported in Figure 9 (full plots in Figure A6). Both referring to the return loss and insertion loss, in Figure 9a and Figure 9b, respectively, the simulated traces predict quite accurately the experiments, despite the fact that an increasing drift is visible at higher frequencies. This leads to a characteristic of the S11 PMD (red trace in Figure 9c) with an approximated constant slope, from nearly 0% to −10%, over the whole range. Instead, the S21 PMD (black trace in Figure 9c) exhibits two error peaks around 6%. The phase characteristic of S11 is missed by the simulation by a large extent, while that of S21 is more accurate. This is visible both looking at the PPD in Figure 9d, as well as confronting the qualitative superimposition of red and black curves in the Smith chart plots (Figure A6q,r).

Eventually, the characteristics of DD in Open and Close state configurations are reported in Figure 10 and in Figure 11, respectively. The full plots are included in Figure A7 and in Figure A8, respectively. Recalling how the layouts of DC and DD compare in Figure 1, it is evident that the concepts are very similar to each other. The only difference is the geometry of the suspended MEMS membrane. In particular, in the case of DC (Figure 1c), it is the classical central square plate, kept suspended by four slender beams and anchored at the other ends. Instead, DD (Figure 1d) is a continuous membrane across the upper and lower anchoring areas, more similar to a bridge. As a consequence, it is reasonable to expect very similar RF characteristics from DC and DD. In fact, the plots in Figure 10 and in Figure 11 replicate very closely those previously discussed and reported in Figure 8 and in Figure 9, respectively. Further comparative considerations are left to the reader, relying on PMD and PPD to ease the interpretation of the simulated and measured S-parameter plots.

In conclusion, bearing in mind the non-trivial challenge of performing RF simulations with unique settings over a frequency range as wide as 110 GHz, the accuracy achieved by the Ansys HFSS 3D models was in general satisfactory to a fair extent. On the other hand, the mentioned non-optimal conditions, along with the sometimes noisy behavior of experiments, provided a spark to develop an articulated discussion. To this purpose, definition and exploitation of PMD and PPD gave a helping hand in quantifying the accuracy of simulations.

It is the author’s belief that such metrics could be useful in the future, whenever there will be the need to elaborate a quantitative comparative discussion among different results. This is considered not only in the case of simulations vs. experiments of RF-MEMS devices, but also when different designs and/or technology solutions must be confronted with each other.

Part of such a scenario will be deployed in the second part of this work, with focus on the detailed comparison of the four RF-MEMS switches performance, along with the modelling of their RF characteristics by means of lumped element networks.

## 4. Comparison of the Four RF-MEMS Switches Performance over Frequency

The target of this section is that of comparing the RF characteristics of the four RF-MEMS switch designs, aiming to infer deductions on how different geometrical features can impact the S-parameters over a frequency range as wide as 110 GHz. Focusing on the Open (O) state configuration, the most relevant parameter to observe is the isolation (S21), reported in the following Figure 12. For brevity, reflection (S11) is omitted here. However, it is reported in Figure A9b, available in Appendix B. This further appendix also places side by side plots collecting all the simulated and measured traces of the four switches. This helps the reader interpret the behavior of S-parameters, as measurements are sometimes quite noisy, and the close-crossing of four characteristics can make it difficult to understand which is which. On the other hand, the cleaner simulated traces are more accessible.

In the lower portion of the frequency range in Figure 12, Device B (DB) scores the best S21 performance up to 45 GHz, being better than around −30 dB up 30 GHz, and better than −12 dB up to 45 GHz. Differently, Device A (DA) scores the worst numbers in terms of isolation, in the range 10–40 GHz. On the other hand, Device C (DC) and Device D (DD) show an intermediate performance, with a rather linear behavior of S21, from −28 dB at 10 GHz to −10 dB at 50 GHz. Above 50 GHz, DA shows better performance in terms of isolation, while DC and DD place intermediately, with S21 always better than −10 dB up to 110 GHz. Finally, DB shows less interesting isolation characteristics.

Focusing on the RF performance in the Close (C) state configuration, return loss (S11) in Figure 13a provides useful indications on the impedance matching over frequency (not developed here for the sake of brevity), while another relevant parameter to observe is the insertion loss (S21) in Figure 13b.

Looking at the whole frequency range in Figure 13b, DA marks the worst performance, confirming the non-viability of this traditional micro-switch design for high frequencies. DB, despite being traditional, shows a more interesting behavior. In particular, S21 worsens down to −3 dB losses at 38 GHz, overlapping quite perfectly with DA. Above 38 GHz, the S21 of DB improves, being better than −3 dB up to 60 GHz. Therefore, it is quite stable around −5 dB up to around 88 GHz, while in the rest of the frequency range it drops to worse values, down to −20 dB at 110 GHz.

Moving now to focus the attention on the more compact design concepts, DC and DD score the rather remarkable performance of having losses (S21) better than −1 dB up to 40 GHz. Above, the return loss remains better than −3 dB up to 50 GHz, and then degrades rather linearly down to −10 dB at 85 GHz. In the higher range, the S21 places around −8 dB up to 110 GHz, scoring a better performance compared to all the other design concepts.

In conclusion, the trend in compacting both the RF-MEMS switching device and the surrounding Coplanar Waveguide (CPW) frame improves quite evidently the S21 performance in the C state configuration. To this end, losses within −1 dB up to 40 GHz are a remarkable result. Nevertheless, further developments will have necessarily to leverage more compact switching membranes, avoiding their transversal placement across the RF central line, exploited in all the configurations discussed above. This would further reduce the parasitic effects introduced by the MEMS membrane in the high frequency range, along with the losses, which would be desirable to keep below −1/−2 dB up to 110 GHz and above. A comparison of the performances achieved by the switches in this work with those of other concepts discussed in literature is included in Table 2.

## 5. Conclusions

The upcoming application paradigms of 6G and Future Networks (FN) will pose unprecedented challenges to Hardware (HW) components, in the years to come. Within such a context, this work focused on RF-MEMS, i.e., Radio Frequency (RF) passives in Microsystems (MEMS) technology, identified as a key enabler of 6G. To this end, four different designs of RF-MEMS series ohmic micro-switches were studied. The RF characteristics (S-parameters) of physical samples were measured up to 110 GHz, and simulated within the Ansys HFSS Finite Element Method (FEM) commercial tool.

In particular, the main contribution of this work can be framed along the following three fundamental directions:Extensive validation of the prediction accuracy achieved by the FEM-based simulation methodology, based on the comparison against Scattering parameter (S-parameters) experimental datasets of the four RF-MEMS designs;Comparison of the RF performances of the four studied design concepts, highlighting pros and cons of each solution, having in mind the emerging needs of 6G and FN;Introduction of ad hoc metrics to help compare in a quantitative fashion the spread of simulations vs. measurements and, more generally, of two S-parameter traces that need to be evaluated against each other.

The discussed tools and methodology, apart from verifying the rather good accuracy of simulations, provided a basis for further discussion and analysis around the suitability of RF-MEMS technology in the 6G/FN scenario. An additional contribution to this topic will be provided in the second part of the work, to be submitted later.

## Figures and Tables

**Figure 1 sensors-23-03380-f001:**
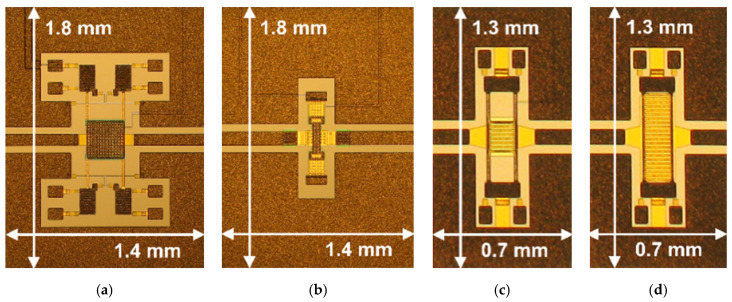
Microphotographs of the RF-MEMS series ohmic switches discussed in this work, along with planar dimensions. The design concepts are named: (**a**) Device A (DA); (**b**) Device B (DB); (**c**) Device C (DC); (**d**) Device D (DD).

**Figure 2 sensors-23-03380-f002:**
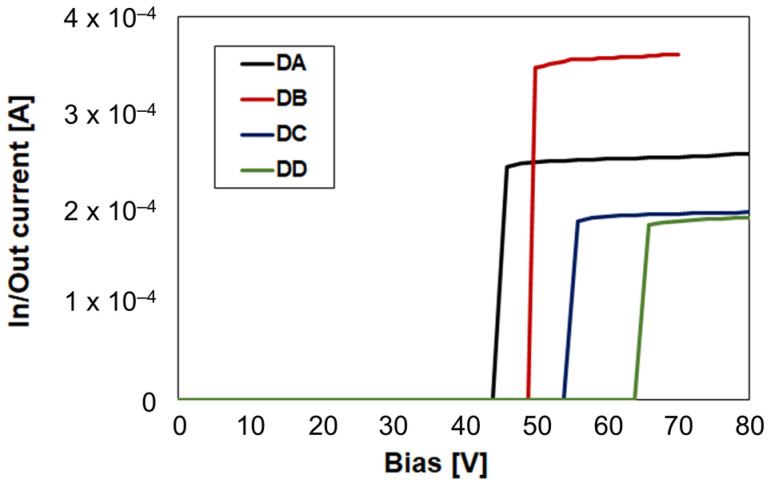
I-V experimental characteristic of the RF-MEMS switches in Figure 1, showing pull-in voltage levels in the 45–65 V range.

**Figure 3 sensors-23-03380-f003:**
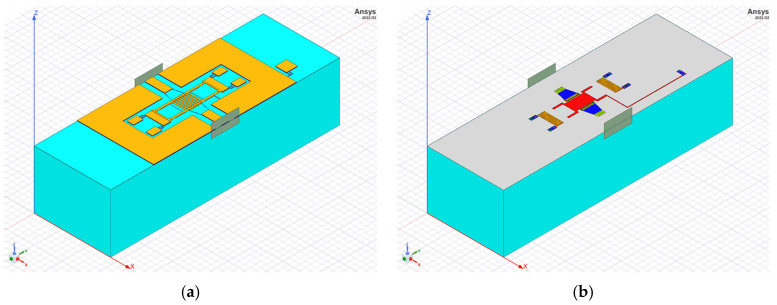
Full-3D Ansys HFSS model of DC in Figure 1c, (**a**) displaying all the layers, (**b**) and hiding the MEMS to display the underneath metallizations.

**Figure 4 sensors-23-03380-f004:**
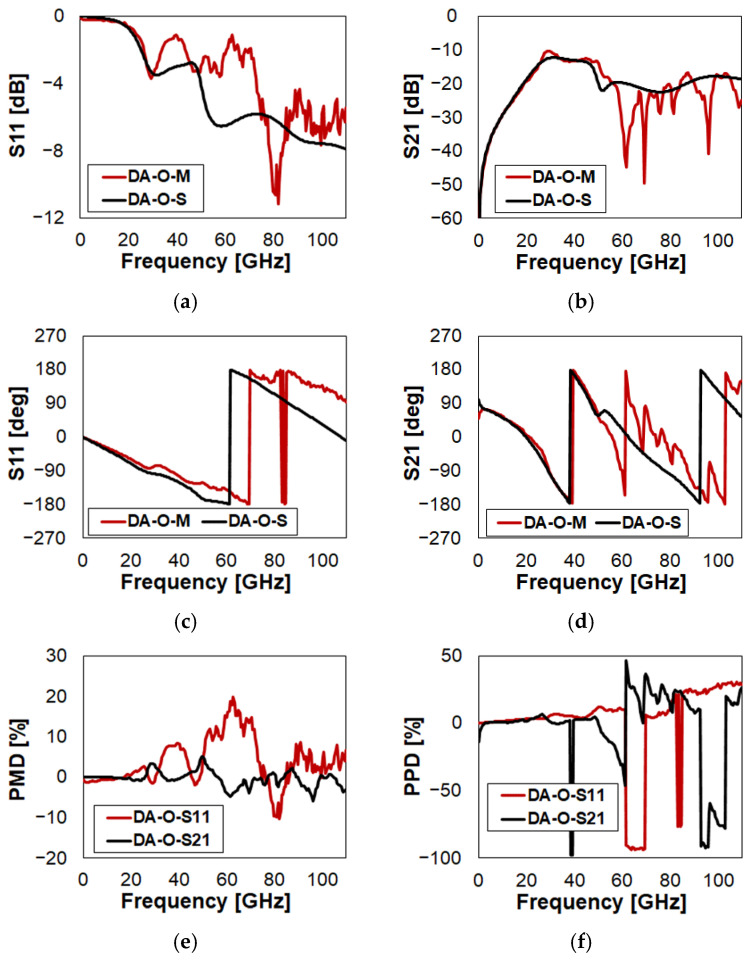
Measured vs. simulated characteristics of Device A (DA) in Open (O) state configuration with reference to: (**a**) reflection (S11); (**b**) isolation (S21); (**c**) phase of S11; (**d**) phase of S21; (**e**) S11/S21 Percent Magnitude Difference (PMD); (**f**) S11/S21 Percent Phase Difference (PPD). Refer to Table 1 for the legends’ nomenclature.

**Figure 5 sensors-23-03380-f005:**
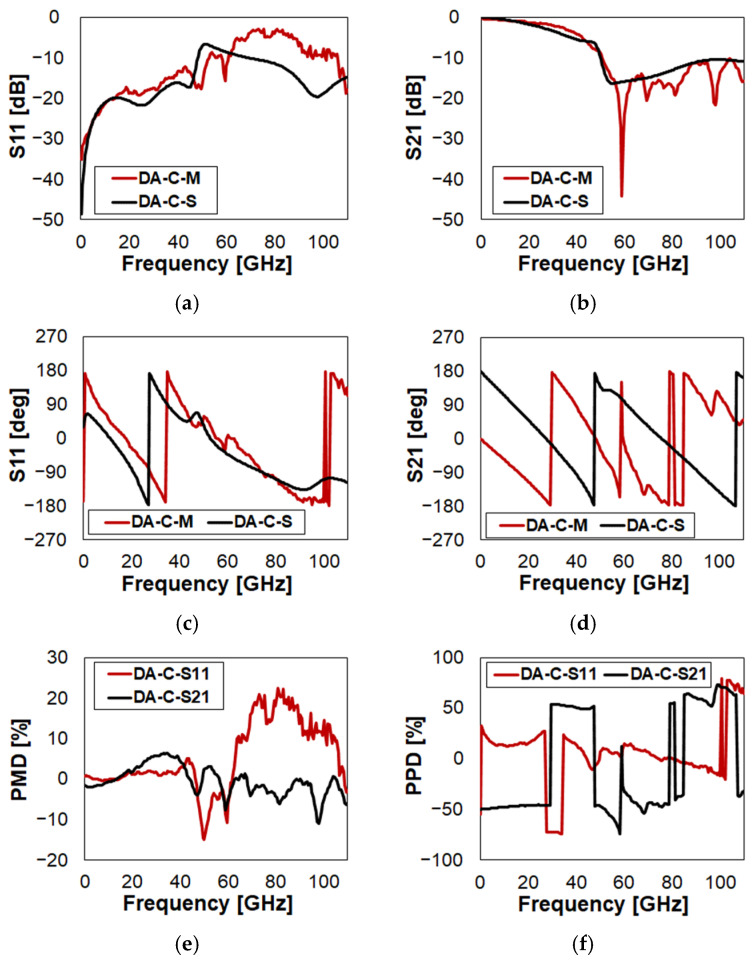
Measured vs. simulated characteristics of DA in Close (C) state configuration with reference to: (**a**) return loss (S11); (**b**) insertion loss (S21); (**c**) phase of S11; (**d**) phase of S21; (**e**) S11/S21 PMD; (**f**) S11/S21 PPD.

**Figure 6 sensors-23-03380-f006:**
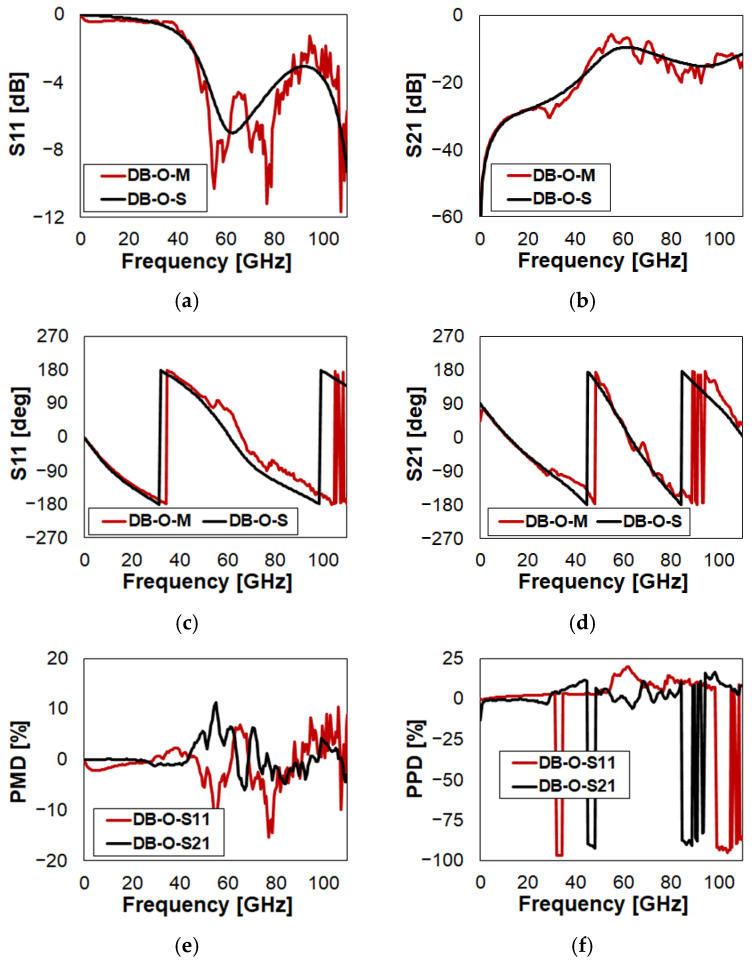
Measured vs. simulated characteristics of DB in O state configuration with reference to: (**a**) reflection (S11); (**b**) isolation (S21); (**c**) phase of S11; (**d**) phase of S21; (**e**) S11/S21 PMD; (**f**) S11/S21 PPD.

**Figure 7 sensors-23-03380-f007:**
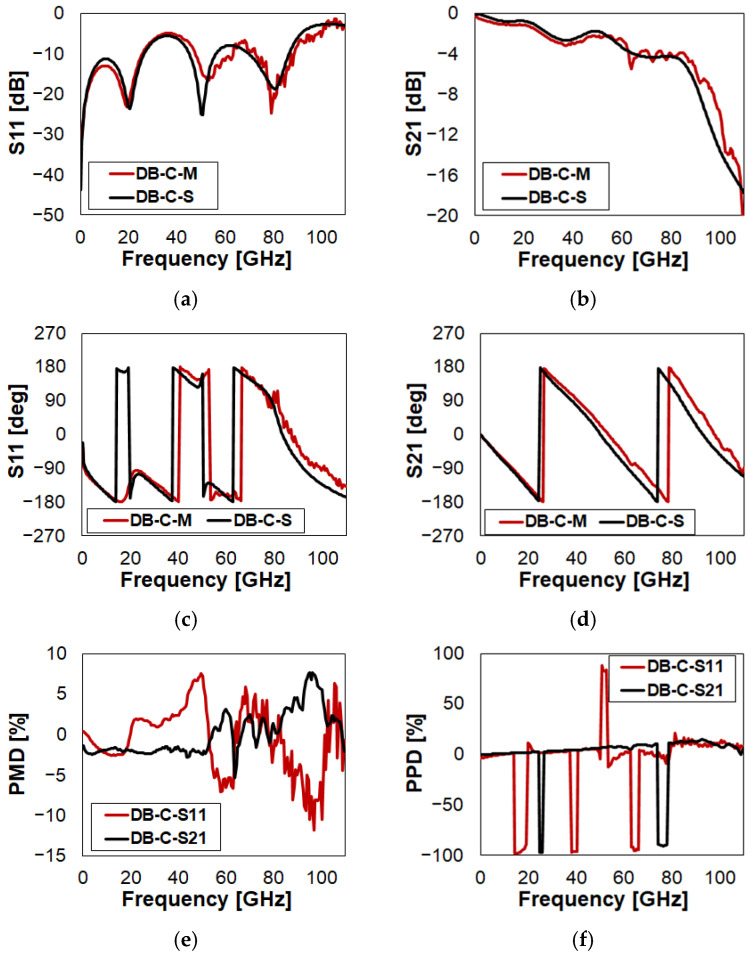
Measured vs. simulated characteristics of DB in Close (C) state configuration with reference to: (**a**) return loss (S11); (**b**) insertion loss (S21); (**c**) phase of S11; (**d**) phase of S21; (**e**) S11/S21 PMD; (**f**) S11/S21 PPD.

**Figure 8 sensors-23-03380-f008:**
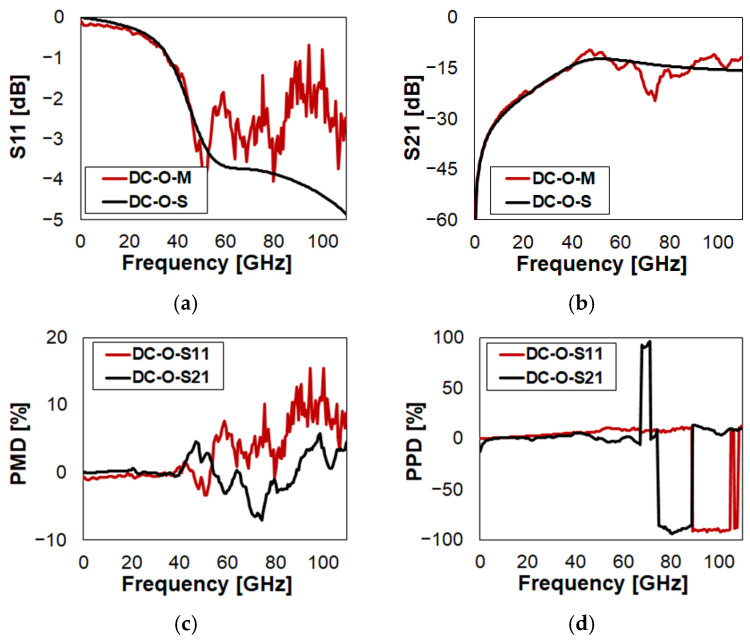
Measured vs. simulated characteristics of DC in O state configuration with reference to: (**a**) reflection (S11); (**b**) isolation (S21); (**c**) S11/S21 PMD; (**d**) S11/S21 PPD.

**Figure 9 sensors-23-03380-f009:**
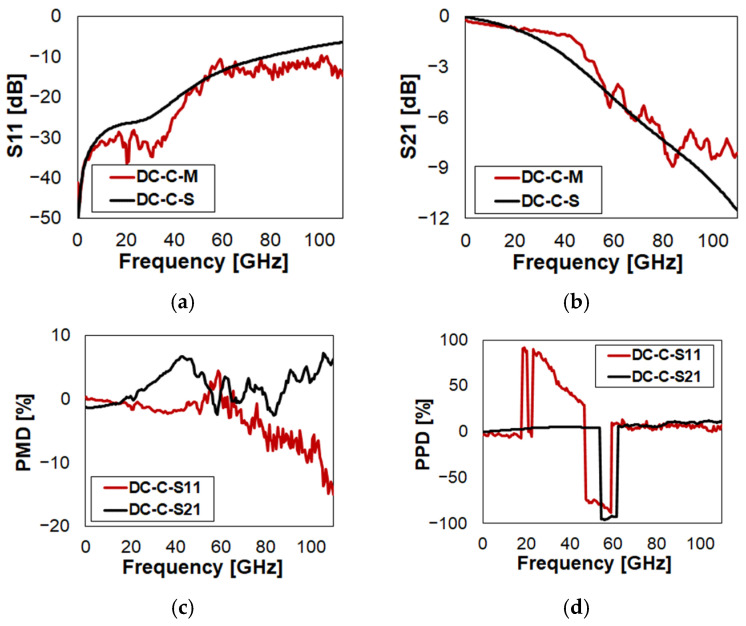
Measured vs. simulated characteristics of DC in Close (C) state configuration with reference to: (**a**) return loss (S11); (**b**) insertion loss (S21); (**c**) S11/S21 PMD; (**d**) S11/S21 PPD.

**Figure 10 sensors-23-03380-f010:**
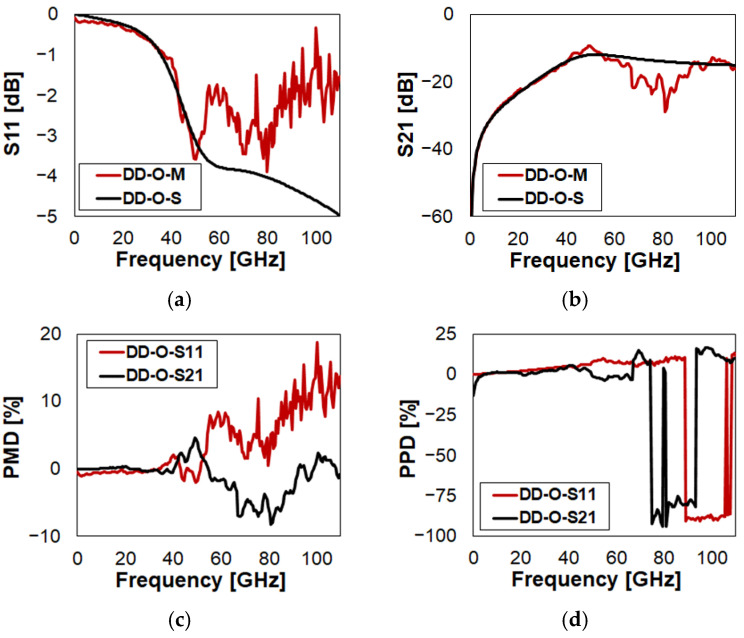
Measured vs. simulated characteristics of DD in O state configuration with reference to: (**a**) reflection (S11); (**b**) isolation (S21); (**c**) S11/S21 PMD; (**d**) S11/S21 PPD.

**Figure 11 sensors-23-03380-f011:**
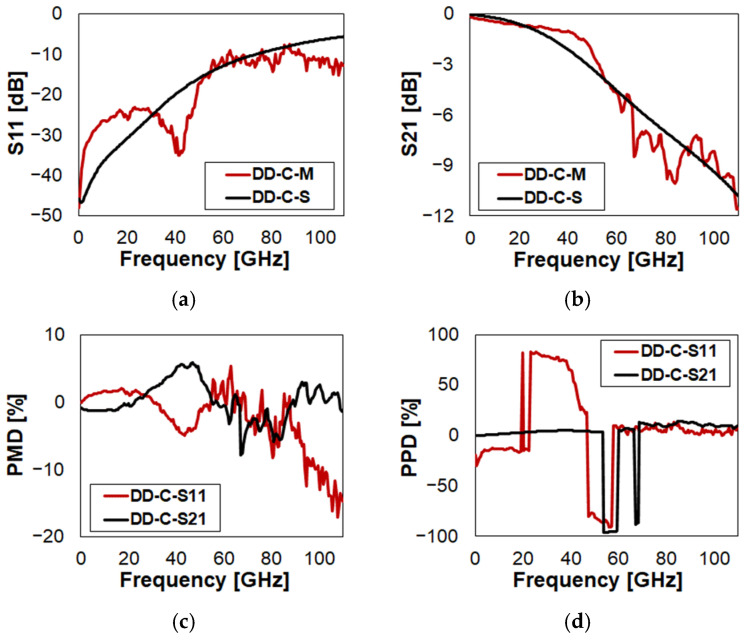
Measured vs. simulated characteristics of DD in Close (C) state configuration with reference to: (**a**) return loss (S11); (**b**) insertion loss (S21); (**c**) S11/S21 PMD; (**d**) S11/S21 PPD.

**Figure 12 sensors-23-03380-f012:**
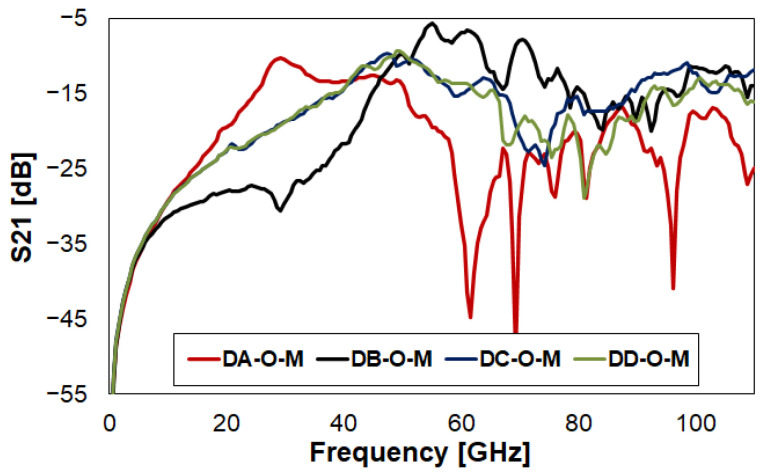
Comparison of the measured isolation (S21) of the four RF-MEMS switches in Figure 1, in Open (O) state configuration.

**Figure 13 sensors-23-03380-f013:**
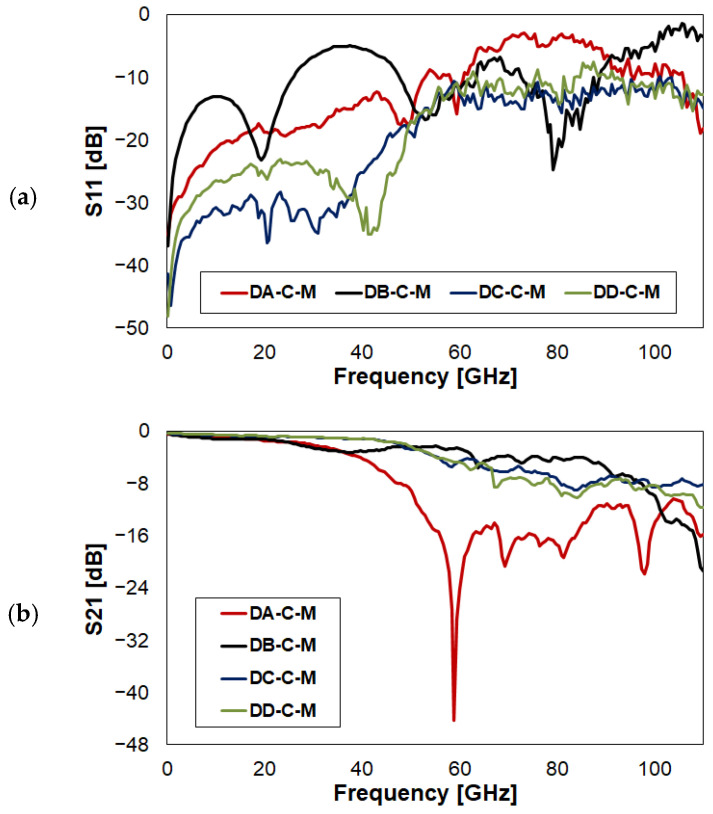
Comparison of the measured characteristics of the four RF-MEMS switches in Figure 1 in Close (C) state configuration with reference to: (**a**) return loss (S11); (**b**) insertion loss (S21).

**Table 1 sensors-23-03380-t001:** Legend taxonomy employed in the following plots.

Acronym	Meaning
DA	Device A
DB	Device B
DC	Device C
DD	Device D
C	Close
O	Open
M	Measurement
S	Simulation
PMD	Percent Magnitude Difference
PPD	Percent Phase Difference

**Table 2 sensors-23-03380-t002:** Comparison of the RF-MEMS switches in this work with other contributions in the literature.

Dev.	Description	Isolation (S21) [Open]	Return Loss (S11) [Close]	Insertion Loss (S21) [Close]
[33]	Radial RF-MEMS switch (electrostatically actuated)	<−8 dB up to 50 GHz	Not reported	>−0.6 dB up to 50 GHz
[34]	Cantilever-type electrostatic switches in a Single Pole 9 Throw structure (simulated results)	−17 dB at 60 GHz	−15 dB at 60 GHz	−0.4 dB at 60 GHz
[35]	Lateral comb-drive electrostatic actuators (bulk micromachining) moving above a CPW on glass substrate	<−27 dB in the range 50–110 GHz	<−17 dB in the range 50–110 GHz	>−2.3 dB in the range 50–110 GHz
[36]	Electrostatic RF-MEMS series ohmic switch (simulated results)	<−21.5 dB in the range 75–110 GHz	<−6.5 dB in the range 75–110 GHz	>−2.4 dB in the range 75–110 GHz
[37]	Electrostatic RF-MEMS shunt capacitive clamped-clamped switch (simulated results)	<−20 dB in the range 20–40 GHz	<−18 dB up to 40 GHz	>−0.3 dB up to 40 GHz
[38]	Lateral comb-drive electrostatic actuators (bulk micromachining) moving above a CPW on glass substrate	<−20 dB in the range 50–110 GHz	Not reported	>−16 dB in the range 50–110 GHz
[39]	Laterally driven RF-MEMS switch actuated by thermoelectric coupling, realized in a surface micromachining process	<−30 dB up to 40 GHz	<−20 dB up to 40 GHz	>−1 dB up to 40 GHz
DA	This work (see Figure 1a)	<−15 dB up to 25 GHz and in the range 50–110 GHz	<−10 dB up to 60 GHz	>−3 dB up to 35 GHz
DB	This work (see Figure 1b)	<−15 dB up to 45 GHz	<−10 dB up to 24 GHz	>−4 dB up to 60 GHz
DC	This work (see Figure 1c)	<−15 dB up to 40 GHz	<−10 dB up to 110 GHz	>−1 dB up to 40 GHz >−3 dB up to 52 GHz
DD	This work (see Figure 1d)	<−15 dB up to 40 GHz	<−10 dB up to 110 GHz	>−1 dB up to 40 GHz >−3 dB up to 52 GHz

## Data Availability

Not applicable.

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
