# Peer review of "Modelling, Validation and Experimental Analysis of Diverse RF-MEMS Ohmic Switch Designs in View of Beyond-5G, 6G and Future Networks—Part 1"

_sensors, 2023, doi:10.3390/s23073380_

Round 1

Reviewer 1 Report

The manuscript is well-written, presenting an exciting and novel work on Radio Frequency (RF) passives in Microsystem (MEMS) technology. The design procedure is clear, and the results are presented professionally. 

I have no comments.

Author Response

The manuscript is well-written, presenting an exciting and novel work on Radio Frequency (RF) passives in Microsystem (MEMS) technology. The design procedure is clear, and the results are presented professionally. 

I have no comments.

== I really thank Reviewer 1 for such a positive opinion and encouraging point of view concerned to the work.

Reviewer 2 Report

four designs of RF-MEMS series ohmic switches are analyzed here. S-parameters (Scattering parameters) are measured and simulated with a Finite Element Method (FEM) tool, in the frequency range from 100 MHz to 110 GHz. A comparative study is then developed around the measured vs. simulated characteristics, also introducing ad-hoc metrics to better quantify the prediction errors and to validate the FEM tool.

1.     Some grammatical errors need to be corrected, and it is recommended to polish the manuscript.

2. In Smith charts, legends need to be aligned

Author Response

four designs of RF-MEMS series ohmic switches are analyzed here. S-parameters (Scattering parameters) are measured and simulated with a Finite Element Method (FEM) tool, in the frequency range from 100 MHz to 110 GHz. A comparative study is then developed around the measured vs. simulated characteristics, also introducing ad-hoc metrics to better quantify the prediction errors and to validate the FEM tool.

== First, I thank Reviewer 2 for carrying out careful check of the whole manuscript.

  1. Some grammatical errors need to be corrected, and it is recommended to polish the manuscript.

== The work underwent significant recheck and polishing, also because of the considerable changes applied.

  1. In Smith charts, legends need to be aligned

== This is certainly a valuable point. However, given the quite scattered behavior of the curves plotted on the Smith charts, leading to diverse coverage of the available polar area, it was found to be more practical keeping the label on a row placed at the top of the chart. By the way, all the Smith chart plots are now removed from the actual article, and left only in Appendix A.

Reviewer 3 Report

I appreciate the efforts from the author in writing such a long manuscript that took me days for reading and thinking. In general this manuscript looks like a technical report rather than a scientific paper or review article. I would suggest the author to clarify the outline and rewrite the manuscript taking into account the following comments.

1. The manuscript should be reduced significantly in length and redundancy, for example, around 10-15 pages for a review article. The abstract should be concise and clear and the redundant background description in the abstract and introduction should be removed.

2. The title looks redundant and confusing. Is this manuscript about an overview of RF-MEMS Ohmic Switches? If yes, the manuscript should include the following information, structural highlights, technical breakthroughs, fabrication process, simulated and measured results, and comparison of/with the state of the art. Please check the manuscript and see if all the contents are well organized and included.

3. The author used numerous acronyms and this would confuse the readers. Again please make the body text concise and clearly readable.

4. The groups from Prof. G. M. Rebeiz, Prof. D. Peroulis, Prof. L. Katehi, etc., are well known for RF MEMS switches, but I am surprised to see that the works from these groups and the RF MEMS book by Rebeiz are missing in this manuscript.

5. Most of the result figures from Figs. 5, 6, 7, 8... look redundant and should be redrawn. Please see the references from Rebeiz and revise the figures.

6. Again, numerous references from the aforementioned groups regarding RF MEMS switches should be included for references and comparison.

Author Response

I appreciate the efforts from the author in writing such a long manuscript that took me days for reading and thinking. In general this manuscript looks like a technical report rather than a scientific paper or review article. I would suggest the author to clarify the outline and rewrite the manuscript taking into account the following comments.

== First, I thank Reviewer 3 for carrying out careful check of the whole manuscript. I understand the concerns and below I report a point by point reply, sketching the changes applied to the manuscript.

  1. The manuscript should be reduced significantly in length and redundancy, for example, around 10-15 pages for a review article. The abstract should be concise and clear and the redundant background description in the abstract and introduction should be removed.

== Several changes were applied to the manuscript. First, the motivation and scopes are now framed more explicitly in the abstract and introduction, identifying three main targets: 1) validation of simulations; 2) performance comparison of different designs; 3) introduction of ad-hoc metrics to help evaluate the spread of different simulated and/or measured characteristics.

== Moreover, some redundant material was removed in the introduction and in the technology description. Also, several plots were removed from the main body of the article, and left just in appendixes.

== On the other hand, the paper did not reduce in length. This is because addressing the above comments required the development of the new Section 4, which goes into details of the performances’ comparison of the four RF-MEMS switches. This additional section also includes a comparative table that puts the performance of the switches here at stake against other relevant contributions in literature.

Also relevantly, this work is envisaged to be the basis for another future contribution from the author. It is also expected to become a valuable basis of data and information for other authors involved in the field of RF-MEMS switches. This is the main motivation for maintaining all the information in the appendixes.

  1. The title looks redundant and confusing. Is this manuscript about an overview of RF-MEMS Ohmic Switches? If yes, the manuscript should include the following information, structural highlights, technical breakthroughs, fabrication process, simulated and measured results, and comparison of/with the state of the art. Please check the manuscript and see if all the contents are well organized and included.

== The title was modified according to this comment. What concerns the organization of the paper is covered by the replies to previous and following points.

  1. The author used numerous acronyms and this would confuse the readers. Again please make the body text concise and clearly readable.

== It is the belief of the authors that acronyms help access information, especially when dealing with plot labels and figure captions. However, some fixing was made, and I hope that now it is easier getting the meaning of the used nomenclature.

  1. The groups from Prof. G. M. Rebeiz, Prof. D. Peroulis, Prof. L. Katehi, etc., are well known for RF MEMS switches, but I am surprised to see that the works from these groups and the RF MEMS book by Rebeiz are missing in this manuscript.

== The mentioned Authors are definitely pivotal in the field of RF-MEMS, and they were not ignored. The point is that the focus of this work is on very-high frequency MEMS devices for 6G and FN applications, while the core of the work of those remarkable Authors is on RF-MEMS in a very general and comprehensive meaning. In any case, a few relevant contributions from the mentioned Authors are now included in the references and grouped all together in the introduction, with the twofold advantage of citing their relevant work, while not making the introduction and state of the art significantly longer.

  1. Most of the result figures from Figs. 5, 6, 7, 8... look redundant and should be redrawn. Please see the references from Rebeiz and revise the figures.

== Significant part of the plots previously displayed in the main text is now removed from the article and left exclusively in Appendixes A and new Appendix B (the latter being linked to the new Section 4).

  1. Again, numerous references from the aforementioned groups regarding RF MEMS switches should be included for references and comparison.

== This comment was addressed in the above reply to remark no. 4.

Reviewer 4 Report

1. Section 2 of this article is titled RF-MEMS switch design concepts, but the first paragraph of the article mainly introduces the manufacturing process. Please describe the similarities and differences of the switch structure in conjunction with Figures 1 and 2, and supplemented in the second paragraph of this section.

Also, is the cross-sectional schematic diagram shown in Fig. 1 consistent with the structural properties of the switch es in Fig. 2? Can the cross-sectional schematic shown in Figure 1 explain and support the fabrication process for the four switches in this paper? It can be seen that the switch
structures shown in FIG. 2 and FIG. 4 are different from the switch structure shown in FIG. 1 , such as the structural characteristics of the upper electrode and the position of the driving electrode.

2. Picture 4 is blurry, please improve the picture clarity.

3. As the article includes qualitative and quantitative analysis, in order to make the data more convincing and accurate, please add appropriate scale lines in Figure 5-12 (excluding the Smith chart) to facilitate data description and observation.

Author Response

  1. Section 2 of this article is titled RF-MEMS switch design concepts, but the first paragraph of the article mainly introduces the manufacturing process. Please describe the similarities and differences of the switch structure in conjunction with Figures 1 and 2, and supplemented in the second paragraph of this section.

== First, I thank Reviewer 4 for carrying out careful check of the whole manuscript. Concerning this point, I believe that a brief description of the technology is necessary, and this is the reason why I decided to maintain it. The discussion on Figure 1 in the previous manuscript version is developed in the following point.

Also, is the cross-sectional schematic diagram shown in Fig. 1 consistent with the structural properties of the switch es in Fig. 2? Can the cross-sectional schematic shown in Figure 1 explain and support the fabrication process for the four switches in this paper? It can be seen that the switch structures shown in FIG. 2 and FIG. 4 are different from the switch structure shown in FIG. 1 , such as the structural characteristics of the upper electrode and the position of the driving electrode.

== Figure 1 in the old manuscript sketches the whole sequence of manufacturing steps as they are in the four RF-MEMS switches, despite it refers to a cantilever-type switching design, rather than to clamped-clamped structures, as Device A to Device D are. This was done because the cantilever-type structure makes possible explaining all the features of the technology within a simple schematic.

However, as the employed technology here is well-known and already widely discussed in previous literature from the author, I decided to remove the schematic from this work, also to meet the need not to make it excessively long, and I included an additional reference to a paper that describes in details the manufacturing process.

  1. Picture 4 is blurry, please improve the picture clarity.

== The 3D schematic were derived once again from the FEM simulator to see if their resolution increased. I hope the current items in Figure 4 look better.

  1. As the article includes qualitative and quantitative analysis, in order to make the data more convincing and accurate, please add appropriate scale lines in Figure 5-12 (excluding the Smith chart) to facilitate data description and observation.

== This is a valuable suggestion, and I tried some solutions in this direction. However, as the sub-plots are very densely placed side by side in the figures, adding scale lines along the XY axes would make them heavier to look at. By the way, a certain amount of plots, also including all the Smith charts, are now removed from the main body of the work, and left exclusively in Appendix A and in the new Appendix B (the latter one linked to the new Section 4). This made the reading and interpretation of sub-plots easier.